# Phase-Homogeneous LiFePO4 Powders with Crystallites Protected by Ferric-Graphite-Graphene Composite

Dmitry Agafonov [1], Aleksandr Bobyl [2,*], Aleksandr Kamzin [2], Alexey Nashchekin [2], Evgeniy Ershenko [2], Arseniy Ushakov [3], Igor Kasatkin [4], Vladimir Levitskii [5], Mikhail Trenikhin [6] and Evgeniy Terukov [7]

1 Department of Electrochemical Production Technology, St. Petersburg State Institute of Technology, Moskovski Ave. 26, 190013 St. Petersburg, Russia
2 Division of Solid State Physics, Ioffe Institute, Politekhnicheskaya Str. 26, 194021 St. Petersburg, Russia
3 Institute of Chemistry, Saratov State University, Astrakhanskaya Str. 83, 410012 Saratov, Russia
4 Research Park, RC XRD, St. Petersburg State University, Universitetskaya nab. 7–9, 199034 St. Petersburg, Russia
5 RnD Center TFTE, Politekhnicheskaya Str. 26, 194021 St. Petersburg, Russia
6 Department "Chemistry and Chemical Technology", Petrochemical Institute, Omsk State Technical University, Mira Ave. 11, 644050 Omsk, Russia
7 Department of Electronics, St. Petersburg State Electrotechnical Univeristy, ul. Professora Popova 5, 197022 St. Petersburg, Russia
* Correspondence: bobyl@theory.ioffe.ru

**Abstract:** Phase-homogeneous LiFePO4 powders have been synthesized. The content of impurity crystalline phases was less than 0.1%, according to synchrotron diffractometry (SXRD) data. Anisotropic crystallite sizes $\overline{L}_{V[hkl]}$ were determined by XRD. A low resistance covering layer of mechanically strong ferric-graphite-graphene composite with impregnated ferric ($Fe^{3+}$) particles < 10 nm in size increases the cycleability compared to industrial cathodes. In accordance with the corrosion model, the destruction of the $Fe^{3+}$-containing protective layer of crystallites predominates at the first stage, and at the second stage Fe escapes into the electrolyte and to the anode. The crystallite size decreases due to amorphization that starts from the surface. The rate capability, $Q(t)$, has been studied as a function of $\overline{L}_{V[hkl]}$, of the correlation coefficients $r_{ik}$ between crystallite sizes, of the Li diffusion coefficient, $D$, and of the electrical relaxation time, $\tau_{el}$. For the test cathode with a thickness of 8 μm, the values of $D = 0.12$ nm$^2$/s, $\tau_{el} = 8$ s were obtained. To predict the dependence $Q(t)$, it is theoretically studied in ranges closest to experimental values: $D = 0.5 \div 0.03$ nm$^2$/s, $\tau_{el} = 8/1$ s, average sizes along [010] $\overline{L}_1 = 90/30$ nm, averaged $\overline{r} = 0/1$.

**Keywords:** energy storage; electrochemical battery; Mössbauer spectroscopy; synchrotron XRD; energy technology; lattice structure; storage degradation; anisotropic crystallite; electrode powder

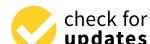



## 1. Introduction

The values of the battery capacity, $Q_0$, the rate capability, $Q(t)$, and the maximum number of discharge-charge cycles are important target battery parameters. As shown in [1], the phase homogeneity of LiFePO4 suffers from the impurity phases which appear as synthesis residues at low temperatures (<550 °C) and ferric ($Fe^{3+}$) compounds at high temperatures in the presence of residual oxygen. The first-principal modelling of Li-Fe-P-O2 phase diagrams [2] supported that.

Several attempts have been made to develop a technology for producing highly efficient and phase-homogeneous electrode powders using: various raw materials and processing methods [3,4], variations in the composition and proportions of the loaded raw materials [5–7], growth duration and additional multi-stage post-growth annealing [8,9], extra pure initial chemicals of so-called "battery quality" [10], one-pot synthesis [11,12], etc. Coating of crystallites with various functional layers was also found useful [13]: metal

oxides, glasses, etc. and the most popular carbon-based materials [14–31] were tried as the coating substances. The latter include various carbon phases [14], nanocarbons of different shapes (spheres, tubes and pores) [15], graphene [16–18]. Sucrose [19–21], glucose [22,23], adipic acid [24–26], polyvinyl alcohol [27,28], polymeric additives [29,30], and ferrocene [31] were used as catalysts for graphitization.

Our approach is based on (i) one-pot LiFePO$_4$ liquid-phase synthesis from high purity lithium and iron acetates taken as the starting materials, (ii) multi-stage thermal processing including acetic acid evaporation at its boiling temperature and protective coating formation from adipic acid and polyvinyl alcohol, (iii) elimination of potential impurity sources, (iv) stage-by-stage control of the effect of the regime parameters on crystallite sizes and coating quality.

X-ray diffractometry (XRD) is commonly used to characterize the presence and size of crystallites in electrode powders. Its synchrotron version (SXRD) [32–34] is the most sensitive to the impurities in crystallites. Various surface-sensitive methods are used to analyze the LiFePO$_4$ particle surface [19,35–38]: Auger electron spectroscopy (AES), electron diffraction (EDS) [39], X-ray photoelectron spectroscopies (XPS) [40–42], inductively coupled plasma (ICP) [39].

We used both synchrotron and laboratory X-ray sources SXRD and XRD, respectively, to increase the sensitivity towards the presence of impurity phases and to determine anisotropic size distributions of LiFePO$_4$ crystallites, as well as to study degradation during cycling of test cathodes.

Mössbauer spectroscopy (MS) makes it possible to detect iron-containing compounds and study their properties [43–49]. In particular, dependence of hyperfine spectral parameters on the charge of Fe ions [45–48] have been used to study reversible delithiation processes in LiFePO$_4$ crystallites. Line broadening associated with an increase in the disorder around Fe$^{2+}$ towards the crystallite surface was observed; oxidation suppression by surface carbon was also found [45]. In this work, the MS results were used for determination of the proportion of the Fe$^{3+}$ ferric impurity compounds in LiFePO$_4$ (which is itself a Fe$^{2+}$ compound).

To reveal the role of ferric compounds, the following results are important: (1) up to 5% ferric states are typically present at regular lattice sites with reduced symmetry [50], which is not detected by XRD; (2) lowering the content of X-ray inactive ferric compounds from 17% to 9% improves the electrochemical parameters, even compared with those materials where the content was zero [51]; (3) presence of significant amounts of Fe$^{3+}$ cations at broad interfaces improves electronic and ionic conductivity [52]; (4) a significant number of ferric states was detected on the surface of crystallites with MS and XPS [53]; (5) MS and ICP studies showed that an increase in the synthesis time led to an increase in the fraction of iron atoms with a long-range order compared to the fraction of iron atoms in the Li/FePO$_4$ amorphous phase [54]; (6) initial powders contained ferric compounds Li$_3$Fe$_2$(PO$_4$)$_3$ and $\alpha$-Fe$_2$O$_3$, which became X-ray inactive after 6 h annealing at 700 °C [55]; (7) after low-temperature precipitation at 106 °C, the powder contained an amorphous ferric phase LiFePO$_4$(OH), which transformed into crystalline LiFePO$_4$ after annealing under reducing conditions; after annealing at 500 °C, the crystallite size increased from 17 nm to 35 nm [56]; (8) with differential scanning calorimetry (DSC) and derivative thermogravimetry (DTG), small particles of Li$_3$Fe$_2$(PO$_4$)$_3$ and presumably of Fe$_2$O$_3$ were detected on the surface of LiFePO$_4$ crystallites after their oxidation below 470–475 °C [57].

We used Mössbauer spectroscopy, EDX and XRD studies to elucidate the non-trivial role of ferric compounds, in particular, in the protective layer formation and its degradation.

The combined use of transmission electron microscopy (TEM) and Raman spectroscopy (RS) is important in study of the various carbon phases in the cathode materials initial powders [58–80]. Using RS, it was found that some carbon materials mixtures have a unique combination of properties: mechanical characteristics, strength, chemical inertness and biocompatibility [61–63]. Among a large number of different grades of carbon black [65], the most famous is Ketjen black [70], which is widely used in electrode powder

technologies [71–76] and is added especially when it is necessary to sharply increase the electronic conductivity [75,76].

The novelty and importance of Raman spectroscopies line intensity analysis lies in the study of the graphene phase's order degree, the graphite phase mechanical strength, and the multilayer graphene phase conductivity compared to industrial $LiFePO_4$ powders. Thus, the development of phase-homogeneous electrode powders may further increase battery capacity and cyclability, and optimize the rate capability.

In this work we synthesized $LiFePO_4$ powders and coatings on the surface of their crystallites. The phase homogeneity was tested with SXRD, and its effect on the test cathodes cyclability was shown. Composition and properties of the layers covering the crystallites was studied using TEM and RS. MS was used to discover the role of ferric compounds, and XRD and EDX were used to study the mechanisms of degradation of the $LiFePO_4$ test cathodes. $LiFePO_4$ test cathodes were subjected to galvanostatic measurements. The hierarchy of relaxation times of electrochemical recharging of the cathodes was determined using the data on anisotropic size distribution of crystallites. We studied the dependence of $Q(t)$ of the test cathodes on the crystallite parameters, the lithium diffusion coefficient D along [010] and the quality of their coating in terms of electric relaxation time $\tau_{el}$. $Q(t)$ calculations allowed us to assess the possibility of improving the technology.

## 2. Synthesis of $LiFePO_4$ Powders and Surface Coatings, and Characterization Techniques

### 2.1. Synthesis of $LiFePO_4$ Powder and Coatings

We used a modified version of the liquid-phase synthesis of $LiFePO_4$ based on lithium and iron acetates as initial reagents [7,81]. They have good solubility in water and low thermal stability at the synthesis temperatures; acetic acid has a low boiling point which ensures its distillation during synthesis, and volatile components can be easily removed from the reaction zone during annealing [82–89]. Initial acetates were prepared from metallic iron and lithium carbonate by interaction with acetic acid (Snabtechmet), grade A.C.S. These materials are available and have a high degree of purity for large-scale production. Chemicals were used as received from the manufacturers; additional reduction of atmospheric impact was provided at the stages of pre-drying and annealing. Mechanochemical activation of the starting materials in a liquid medium promoted formation of an intermediate synthesis product, which was converted into $LiFePO_4$ at a temperature lower by 100 °C than in the standard procedures. This created the prerequisites for obtaining $LiFePO_4$ with a reduced content of impurity phases [87]. Mechanochemical activation was carried out in a saturated solution of ammonium dihydroorthophosphate in distilled water. Table 1 shows the following synthesis steps:

**Table 1.** Sequence of $LiFePO_4$ liquid-phase synthesis.

| № | Steps | Reagents |
|---|---|---|
| 1 | Preparation of acetates | $Fe + LiCO_3 + CH_3COOH \rightarrow$ $Fe(CH_3COO)_2, LiCH_3COO, H_2O$ |
| 2 | Organic Additives | AA, PA |
| 3 | Phosphoric acid | $H_3PO_4$ |
| 4 | Pre-drying at 100 °C | Evaporation of $CH_3COOH, H_2O$ |
| 5 | Annealing 1.5 h at 400 °C in an Ar atmosphere | Evaporation of $CH_3COOH, H_2O, CO_2,$ $((CH_2))_4CO$ |
| 6 | Annealing 1.5 h at 670 °C in an Ar atmosphere | Crystallization of $LiFePO_4$ |

1. Lithium acetate is obtained by direct acetic acid action on lithium carbonate. The $Fe^{2+}$ ion is easily oxidized by atmospheric oxygen to $Fe^{3+}$, therefore, iron acetate is prepared by placing a calculated amount of iron in a flask containing an excess of acetic acid. The flask is stoppered and kept under vacuum until dissolved with stirring with a magnetic stirrer and heated at the end of the process.

2. Two organic additives, adipic acid (AA) [88–91] and polyvinyl alcohol (PA) [92–94], are used to obtain and condition the synthesized composite. AA crystallizes upon reaction mixture evaporation and decomposes upon heat treatment, resulting in a precipitate with a loosened structure. PA is adsorbed on the precipitate surface and contributes to the small crystals formed during the interaction of phosphoric acid and lithium and iron acetates.

3. Phosphoric acid is used as a source of phosphorus ions.

4. Pre-drying to evaporate $CH_3COOH$ and $H_2O$ in a stream of hot air, which prevented agglomeration of the starting materials.

5–6. Annealing for evaporation at different temperatures and crystallization. After drying in a stream of hot air, the sample was pelletized and placed in a sealed muffle furnace, which was constantly purged with especially pure nitrogen. The furnace was heated to 400 °C, the sample was held at this temperature for 1.5 h, then the heating was turned off and the furnace was expected to cool in a nitrogen flow to a temperature of 25 °C. The resulting intermediate was subjected to repeated grinding and tableting. This approach does not lead to the formation of the final product due to the low temperature, but it makes it possible to obtain a dense mixture of initial substances with a large interfacial surface and a fixed contact between the phases of the initial substances. Then the tableted sample is placed in a sealed muffle furnace, which is constantly purged with especially pure nitrogen; the temperature in the furnace rises to 670 °C. This approach is tested in a series of experiments near this temperature. The selected temperature regime provides the maximum $LiFePO_4$ capacity of 65 mAh/g at 20 C rate. At all stages of preparation and synthesis, the above measures are taken, in particular, to exclude the transition of $Fe^{2+}$ to $Fe^{3+}$.

Preliminary technological experiments were also carried out, in particular using: iron acetate $(CH_3COO)_2Fe$, iron oxalate $FeC_2O_4 \times 2H_2O$, lithium carbonate $Li_2CO_3$, ammonium dihydroorthophosphate $(NH_4H_2PO_4)$ from VEKTON, grade A.C.S. Depending on the composition and the annealing modes, the following impurity crystalline phases were observed in the samples: $Fe_3O_4$ (ICDD 00-019-0629), $Fe_2O_3$ (ICDD 00-039-1346), $FeCO_3$ (ICDD 00-029-0696), $FePO_4$ (ICDD 00-050-1635). To compare the quality and target parameters of the developed sample N1, SPbTU, the following industrial $LiFePO_4$ powders were used: N2, Phostech Lithium [95]; N3 OCELL Technologies N4, Golden Light Energy. The powders had specific capacities ranging from 145 mAh/g to 167 mAh/g at 0.1 C rate [7,81], given in the Table 2 at 20 C rate.

**Table 2.** SXRD composition of impurity crystallites in powders [96], XRD anisotropic sizes of $LiFePO_4$ crystallites $\overline{L}_{V[hkl]}$ averaged over the length of their columns [97] (from [98]). MS fractions of $Fe^{2+}$ and $Fe^{3+}$ compounds (description below). Forecast cycling of a test cathode after 100 times cycling at 1 C rate. The errors of $\overline{L}_{V[hkl]}$ reported in parentheses characterize the reproducibility.

| | Impurity Phase, % | $Q$, mAh/g 20 C Rate | $\overline{L}_{V[100]}$, nm | $\overline{L}_{V[010]}$, nm | $\overline{L}_{V[001]}$, nm | Forecast Cycling | Mössbauer $Fe^{2+}$, % | $Fe^{3+}$, % |
|---|---|---|---|---|---|---|---|---|
| N1 | not detected | 63 | 66 (5) | 82 (5) | 89 (7) | 3500 | 96 | 4 |
| N2 | $Li_3PO_4$, 1.01 (2) | 58 | 145 (26) | 131 (13) | 185 (17) | 1000 | 95 | 5 |
| N3 | $Li_3PO_4$, 2.39 (3) $Fe_2P$, 2.34 (3) $Fe_3P$, 2.02 (3) | 57 | 141 (5) | 146 (15) | 165 (7) | 5000 | 92 | 8 |
| N4 | not detected | 40 | 230 (20) | 261 (8) | 242 (30) | 800 | 98 | 2 |

The novelty of our synthesis in comparison with [87–89] is the following set:

1. One-pot $LiFePO_4$ liquid-phase synthesis using chemically pure lithium and iron acetates as starting materials.

2. Using raw materials of organic nature.

3. Using the tableting operation after the pyrolysis of organic matter, at 400 °C. This operation facilitates the course of the final synthesis of the topochemical reaction.
4. Drying, before heat treatment, was carried out in a stream of hot air, which prevented the agglomeration of the starting materials.
5. The synthesis intermediate is tableted, heat-treated at 400 °C, then re-milled and re-tableted before the final heat treatment at 670 °C. This made it possible to obtain a material of high phase purity.
6. Before repeated tableting, adipic acid was introduced, the pyrolysis of which in an inert medium (high-purity nitrogen) led to encapsulation of $LiFePO_4$ in a carbon shell.
7. At all stages of the preparation, measures were taken to exclude the transition of $Fe^{2+}$ to $Fe^{3+}$, in particular, by isolation from atmospheric moisture.

### 2.2. Characterization Techniques

Note that conventional XRD is less sensitive than SXRD due to the lower intensity of laboratory radiation sources [32–34,96]. The SXRD experiments were done at the Structural Materials Science station of the Siberia-2 synchrotron radiation source of Kurchatov Institute Research Center [96]. Measurements were made at a wavelength of λ = 0.68886 Å in the transmission (Debye–Scherrer) mode with a Fujifilm Imaging Plate memory layer as a 2D detector, sample-to-detector distance of 200 mm, and exposure time of 15 min.

The anisotropic crystallite sizes $\overline{L}_{V[hkl]}$ averaged over the length of the columns [97] were determined by XRD [98]. Bruker D8 Discover diffractometer was used in a parallel-beam linear-focus mode at $2\theta$ = 15–125 deg, and MAUD software was utilized for profile fitting [94]. The primary beam was conditioned with a double-bounce channel-cut Ge220 monochromator to provide $CuKa_1$ radiation with a wavelength of 1.54056 Å. The Cagliotti coefficients of the instrumental profile function were refined by fitting the data for a $LaB_6$ powder specimen (NIST SRM 660c).

The powders were studied with a JEOL JEM 2100 high-resolution TEM at an accelerating voltage of 200 kV and crystal lattice resolution of 0.14 nm. The instrument was equipped with an INKA 250 Xray spectrometer. An image of the single-crystal gold lattice with the (111) interplanar spacing of 0.235 nm was used as reference for linear scale calibration.

Raman spectra were measured at room temperature in the "backscattering" geometry on a LabRam HR 800 spectrometer equipped with a confocal microscope. The measurements used the exciting light wavelengths of 532 nm and 633 nm, focused on the surface of the sample into a spot with a diameter of ~1 μm. In this case, the laser radiation power on the sample was maintained at a level of 2.0 mW. The use of a 600 pcs/mm diffraction grating made it possible to obtain a spectral resolution no worse than 2.5 cm$^{-1}$.

The Mössbauer effect was measured at room temperature on 57Fe nuclei in the γ-ray transmission geometry through powders sputtered onto aluminum foil with a spot diameter of 20 mm. The movement of the 57Co(Rd) γ-radiation source in the spectrometer was carried out with a constant acceleration of the reference signal in the form of a triangle. Velocity calibration was performed using α-iron foil for two Doppler shift velocities of the gamma-ray source.

### 3. $LiFePO_4$ Powders Phase Homogeneity Studies Using SXRD and Its Effect on the Test Cathodes Cyclability

#### 3.1. Phase Homogeneity Using SXRD

The results are shown in Figure 1 and the quantitative composition of impurity crystalline phases are given in Table 2.

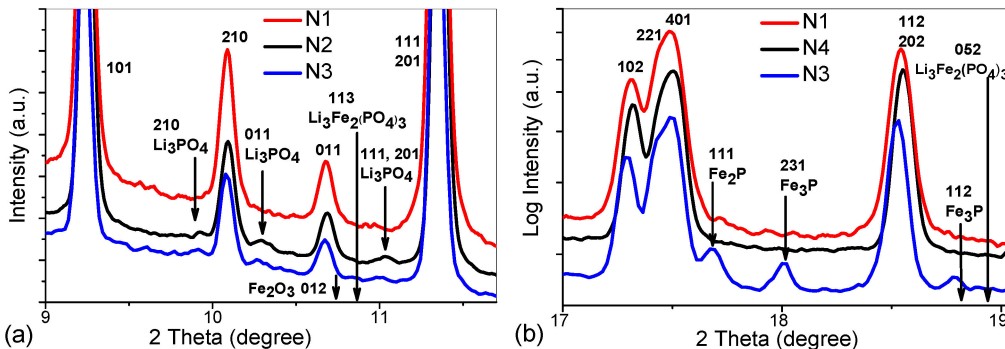

**Figure 1.** SXRD measurements of the developed and industrial LiFePO$_4$ powders, N1 and N2–4, respectively, for 2 Theta intervals 9.0–11.7 (**a**) and 17.0–19.1 (**b**). Arrows near the abscissa indicate the expected peak positions.

### 3.2. Cycling Test Cathode Cells

A test electrode was prepared of 80 wt.% powder, 10 wt.% acetylene black and 10 wt.% polyvinylidene fluoride (PVDF). It was applied as a sample homogenized suspension of acetylene black in a 5 wt.% solution of PVDF in N- methylpyrrolidone (analytical grade) on 1 cm$^2$ aluminum plate, 0.4 mm thick, after which it was dried at 120 °C in air for 12 h [99]. To reduce the errors of the galvanostatic measurements of rate capability at small times, the cathode thickness was minimal, about 8 μm. The measurements included charge-discharge cycles with constant current loads: sequentially from 0.1 to 20 C rates in one cycle, then 15 cycles each, completing 10 cycles with a load of 1 rate. The current density of 1 C rate corresponded to 170 μA per 1 mg of sample and was analyzed at the potentials in the range of 2.6–4.3 V relative to the lithium reference electrode. Figure 2 shows the galvanostatic measurements results. Cycling predictions were obtained by making 150 charge-discharge cycles and by recalculating to the point of the capacitance reduction to 80 % of its initial value at a current of 1 C rate indicated in Table 2.

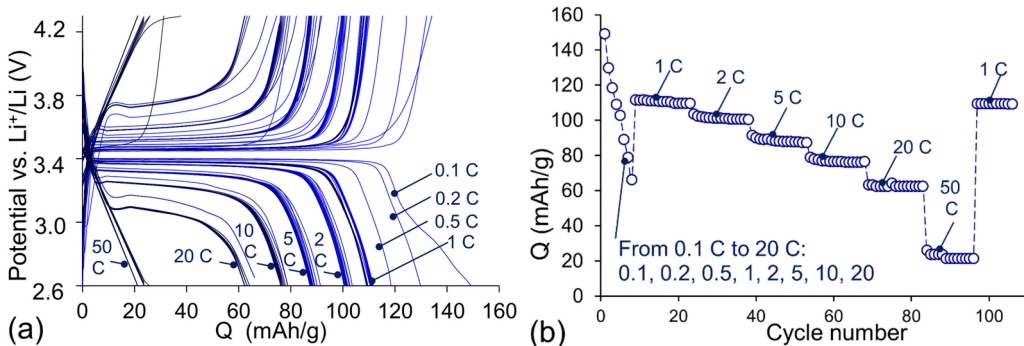

**Figure 2.** (**a**) Galvanostatic charge-discharge curves, (**b**) variation of capacity with the cycle number. The values of the charge and discharge currents (C rate -units) are on the diagrams.

The following conclusions can be drawn from the results in Table 2:

1. No impurity crystallites were found in the samples N1 and N4. However, the cyclability of the latter was significantly worse than that of the developed sample N1, despite the trend towards an increase in the role of the [100] surface of larger crystallites in their cyclability. Consequently, the absence of impurity crystallites is a necessary but obviously not a sufficient condition.

2. Sample N1 had a high cyclability, but lower than that in the sample N3, which contained several impurity phases. In addition to lowering the resistance, these phases probably catch the degradation products, which slows down the formation of harmful impurity phases. It can be concluded that it is promising to search for such

compositions or isovalent doping, including various mixed solid solutions, starting from our pure technology.

## 4. Crystallite Coating Composition and Properties Studied with TEM and RS

### 4.1. TEM Study

TEM images of LiFePO$_4$ powders are shown in Figure 3. Several types of particles are observed in the powders:

-   Large 40–150 nm particles of LiFePO$_4$ are observed in all TEM images; an example is shown in Figure 3a for the developed sample N1.
-   In almost all samples, nanocrystalline 5–10 nm Li$_3$Fe$_2$(PO$_4$)$_3$ particles are observed (Figure 3b,e,f) with lattice spacing of 0.428 nm, which corresponds to (200) or (−121) planes. It should be noted, however, that the (011) planes of LiFePO$_4$ have a similar interplanar distance. The ferric compound Li$_3$Fe$_2$(PO$_4$)$_3$ on the surface of LiFePO$_4$ crystallites appears as a result of insufficient oxygen content to complete the oxidation reaction.
-   Figure 3f shows crystallites with interplanar spacing of 0.220 nm that corresponds presumably to the (321) planes of Fe$_3$P in the N3 sample, even though a similar distance can be found in the structures of other phases. According to [91,101,102], at T > 850 °C and in the presence of carbon, LiFePO$_4$ is reduced to form Fe$_3$P. As seen from Table 2, a significant amount of that phase is reliably detected in the sample N3 using SXRD; the plate-like shape of the particles was described in detail in [103].
-   Various structures of the carbon layers encapsulating the LiFePO$_4$ particles can be seen in the samples. More ordered carbon shells are up to 5 nm thick and the amorphous shells are up to 20 nm thick. In some cases, particles without a carbon shell are observed. The properties of the carbon coatings will be discussed below in the RS section.

In none of the samples have particles of Fe2O3 been found. Perhaps they can be observed in other technologies, considering also some identification uncertainty with Li$_3$Fe$_2$(PO$_4$)$_3$, so the coating we observed was called a ferric-graphite-graphene composite.

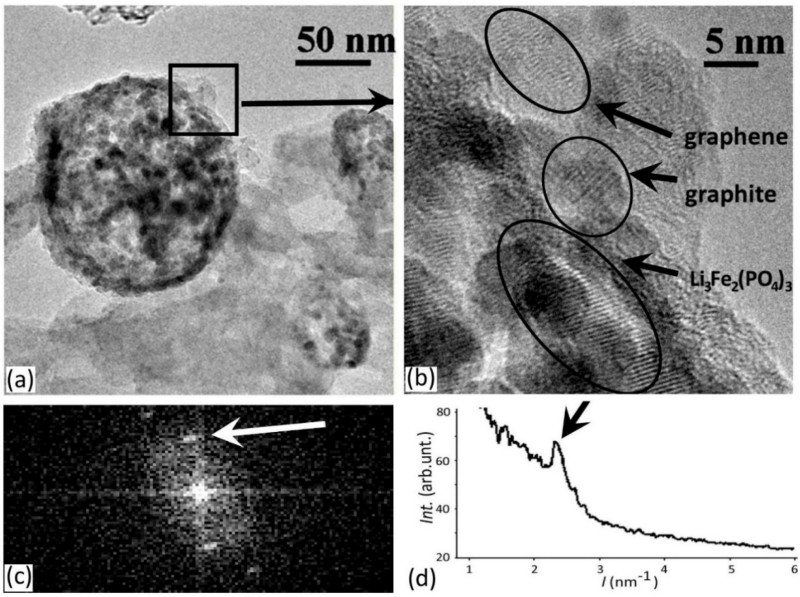

**Figure 3.** *Cont.*

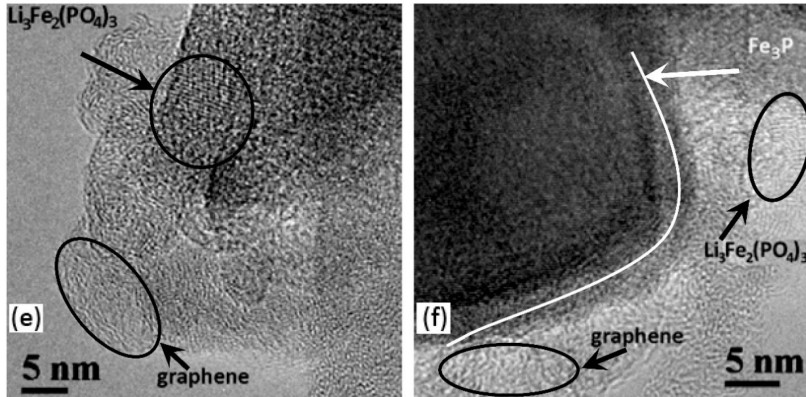

**Figure 3.** (**a**–**d**)—TEM images of the developed N1 powder of 3 crystallites (**a**); (**b**) is an enlarged part of (**a**). Areas of the ferric-graphite-graphene composite coating layer are marked (**b**); FT snapshot obtained from this TEM image (**c**), a maximum close to the interplanar spacing of 0.428 nm (**d**). (**e**,**f**)—TEM images of industrial powders: N4 (**e**) and N3 (**f**) containing the impurity crystallite phase of $Fe_3P$. Regions characteristic of multilayer graphene and the most probable $Li_3Fe_2(PO_4)_3$ are also marked.

### 4.2. Raman Spectroscopy Studies of Carbon Phases

Figure 4 shows the Raman spectra of the Sample N1 taken in the ranges from 800 to 3700 cm$^{-1}$ [104].

**Table 3.** The Raman spectra parameters: E1, I1 and W1—position of the maximum, area and width of one Lorentz approximation for the 1st peak and so on for the others shown in Figure 4 for the excitation wavelengths of 532 nm and 633 nm.

| | E1, I1, W1 | E2, I2, W2 | E3, I3, W3 | E4, I4, W4 | E5, I5, W5 | E6, I6, W6 | E7, I7, W7 | ID/IG (I2/I4) | Isp2/Isp3 ((I2 + I4)/(I1 + I3)) | (I1 + ... AI)/ (I5 + ... I7) |
|---|---|---|---|---|---|---|---|---|---|---|
| | | | | Excitation 532 nm | | | | | | |
| N1 | 1203 64,354 130 | 1346 464,462 191 | 1513 109,518 128 | 1596 183,271 64 | 2681 84,869 391 | 2915 114,677 376 | 3176 7613 112 | 2.53 | 3.73 | 3.96 |
| N4 | 1187 45,591 94 | 1341 733,854 190 | 1527 130,196 119 | 1599 244,016 58 | 2668 146,229 337 | 2919 231,601 344 | 3180 17,221 113 | 3.01 | 5.56 | 2.92 |
| N2 | 1190 22,138 110 | 1342 288,779 188 | 1522 53,844 120 | 1600 98,169 59 | 2651 37,209 307 | 2909 88,666 380 | 3185 5240 117 | 2.94 | 5.09 | 3.53 |
| | | | | Excitation 633 nm | | | | | | |
| N1 | 1201 63,280 131 | 1346 475,058 195 | 1514 106,974 126 | 1596 182,032 64 | 2716 65,290 328 | 2949 38,599 234 | 3178 9359 128 | 2.61 | 3.86 | 7.3 |
| N4 | 1193 11,389 112 | 1332 133,742 180 | 1528 21,170 123 | 1602 37,709 56 | 2630 16,408 293 | 2883 20,492 318 | 3178 692 78 | 3.55 | 5.26 | 5.42 |
| N2 | 1198 7329 117 | 1331 69,319 179 | 1516 11,644 127 | 1601 19,955 57 | 2594 9304 340 | 2870 9744 334 | 3178 350 130 | 3.47 | 4.70 | 5.58 |

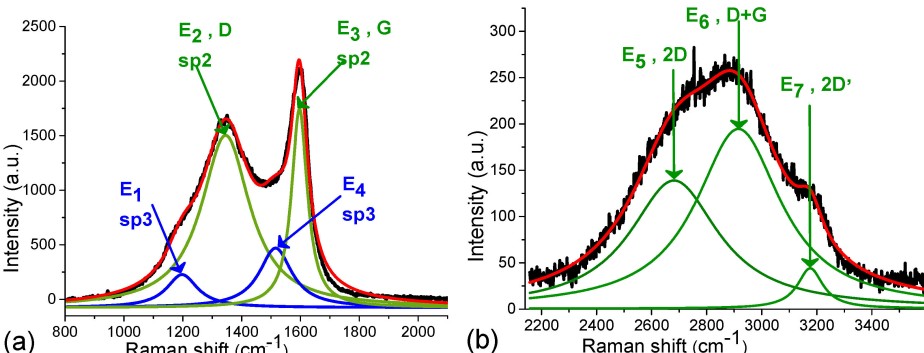

**Figure 4.** Sample N1 Raman spectra at 532 nm in the range 800–2100 cm$^{-1}$ (**a**) and second ordered RS in the range 1900–3700 cm$^{-1}$ (**b**). Decomposition into the Lorentz components with energies in their maxima E1–E7, given in Table 3, is shown.

The RS spectrum analysis is performed by the component separation method using a Lorentz line shape [105,106]. To improve reliability, two close excitation lines 532 nm and 633 nm are also used, since it is expected that the main conclusions from the results of comparing the line amplitudes should coincide in two measurement series. Table 3 includes all decomposition parameters useful for comparison between this and other studies, especially the second order linewidths.

According to [58–60], two lines at 1518 cm$^{-1}$ and 1201 cm$^{-1}$ have already been observed in disordered carbon black and diamond-like carbons. That could imply that the short-range vibrations of the sp3-coordinated carbons contribute to the disordered spectra. Unique sp2/sp3 nanohybrids as bulky nanodiamonds (NDs) and sp2 concentric onion-like carbons (OLC) [61,62] with outstanding mechanical performance, sufficient chemical inertness, excellent biocompatibility, high mechanical strength [61–63] are possible.

To interpret the lines in Figure 4, we use the sequence proposed in [64]. The spectra show lines arising from light scattering in spectral regions close to their position in the spectra of multilayer graphene or graphene-like layers: lines D (E2), G (E4), 2D (E5), D + G (E6), 2D′ (E7) [64–66]. Line G in the spectrum of graphene corresponds to nonresonant light scattering involving an optical phonon of E2g symmetry with a small wave vector. This phonon is caused by vibrations of carbon atoms in the layer plane. The appearance of the D line in the spectra is explained by resonant scattering involving electronic states from two nonequivalent K- and K′-points of the Brillouin zone and an optical phonon with a large wave vector. This process is forbidden by the quasi-momentum selection rule, but the condition for its conservation can be satisfied if the crystal lattice defect also participates in the scattering process. In structurally perfect graphene samples, line D should not be observed. The second order spectra in Figure 4b are markedly broadened. The nature of the 2D line, an overtone of the D line, is also associated with resonant light scattering involving electronic states. The quasi-momentum conservation conditions for such a process are always satisfied, so the 2D line will be present in the graphene spectrum even if it does not contain the D line. The combination D + G corresponds to a defect-induced double resonance "inter-valley" scattering process which is allowed through a defect-induced triple resonance process. The second order spectra lines are observed in graphene oxide, GO, [67,68], which is produced through graphite chemical oxidation and subsequent exfoliation via sonication, and in various modifications of carbon black (CB) powders [69].

It should be noted that the positions and widths of the observed second ordered three lines in the range 1900–3700 cm$^{-1}$ correspond to those described in [70–72,77–79]. In particular, the papers [74,79] describe the dependence of the position of the 2D line, with a maximum of E5, on the magnitude of the deformation. At the same time, its width in Figure 4b is almost 10 times larger than that described in [64–66] and is comparable to its displacement at deformations of about 1%, which indicates the presence of a significant deformation disorder in the studied samples compared to multilayer graphene.

The carbon fraction diagnostics using Raman spectroscopy data is based on an analysis of the position, the width, and the intensity ratio of the observed lines. In our case, the integral intensity is obtained from the results of decomposition into components. The best known are the intensity ratios ID/IG and $Isp^2/Isp^3$ [64–66], in the used designations I2/I4 and (I2 + I4)/(I1 + I3), respectively. Obviously, the ratios of the total line intensities of the first and second order can also be useful, i.e., (I1 + . . . I4)/(I5 + . . . I7) [68]. Comparing the ratios of the line intensities, we can draw the following conclusions:

For the developed sample, the ID/IG (I2/I4) ratio was minimal compared to the industrial control samples. This indicates a smaller amount of disorder in the multilayer graphene subsystem and smaller sizes of clusters in their amorphous part [64–66].

The $Isp^2/Isp^3$ ((I2 + I4)/(I1 + I3)) ratio was also minimal in comparison with the control industrial samples. This means a larger role of graphite short-range sp3 bonds in the mechanically stronger amorphous part compared to the graphene multilayer lobes and the bridges between crystallites [58–60].

The ratio (I1 + . . . AI)/(I5 + . . . I7) was at a maximum suggesting a higher conductivity of the multilayer graphene due to a higher degree of screening of overtone photon-phonon interactions by carriers [64–66,68].

## 5. The Role of Ferric Compounds Studied with MS; Degradation Mechanisms of LiFePO$_4$ Test Cathodes with XRD

### 5.1. Mössbauer Spectroscopy Studies

The obtained Mössbauer spectra are processed by the least squares method using the Lamb–Mössbauer factors [107]. Table 4 shows the MS calculations results obtained with a high Doppler shift rate of the gamma source. Figure 5 shows the Mössbauer spectra of sample N1, and Table 4 shows the calculation results. From Figure 5 and Table 4, it can be seen that the Mössbauer spectra consist of two doublets superimposed on each other, and no additional lines indicating the presence of another phase are observed. The values of the $Fe^{2+}$ and $Fe^{3+}$ absorption lines relative intensities are determined from the experimental spectra; the line half-widths are given in Table 4.

**Table 4.** Parameters of hyperfine interactions obtained by mathematical processing of Mössbauer spectra.

| Line Marking | IS, mm/s | QS, mm/s | G, mm/s | Int, (%) | Charge State Fe |
|---|---|---|---|---|---|
| $Fe^{2+}$ | $0.981 \pm 0.001$ | $2.928 \pm 0.001$ | $0.281 \pm 0.001$ | $94.8 \pm 0.2$ | $Fe^{2+}$ |
| 1 | $1.156 \pm 0.040$ | $1.818 \pm 0.028$ | $0.345 \pm 0.025$ | $1.4. \pm 0.3$ | $Fe^{2+}$ |
| 2 | $0.172 \pm 0.080$ | $0.830 \pm 0.080$ | $0.345 \pm 0.025$ | $2.6 \pm 0.4$ | $Fe^{3+}$ |
| 3 | $0.404 \pm 0.400$ | $0.452 \pm 0.060$ | $0.345 \pm 0.025$ | $1.2. \pm 0.3$ | $Fe^{3+}$ |

As can be seen in Figure 5, MS at room temperature (295 K) does not show any magnetic ordering lines traces, and the spectrum consists of quadrupole doublets. This means that the sample is in a paramagnetic state and no magnetic ordering traces or relaxation processes are observed. High intensity doublet lines are symmetrical. The spectra show a doublet with a small linewidth, maximum intensity, and hyperfine interaction (HFI) parameters: IS = 0.981(1) mm/s, QS = 2.926(2) mm/s. A doublet with similar HFI parameters IS = 1.23 mm/s and QS = 2.96 mm/s is observed for iron ions in the olivine structure, which corresponds to the high-spin iron $Fe^{2+}$ in an octahedral M2 environment [48,80,108]. In the case of LiFePO$_4$ with the olivine structure, a doublet with IS = 1.22 mm/s and QS = 2.80 mm/s, which is attributed to $Fe^{2+}$ ions, is also observed [45,46]. However, the IS values obtained from experimental MS (Figure 5) are somewhat lower (0.981 mm/s). The small linewidth of the dominant doublet (0.281 mm/s) means that the $Fe^{2+}$ ions occupy positions in the well-ordered LiFePO$_4$ phase structure. The IS and QS values are close to those obtained using the density functional theory (DFT) calculations, taking into account both the spin polarization and the correlation of Fe 3d electrons [109]. This can be explained

by the high-spin configuration of $Fe^{2+}$ ions in a distorted octahedral environment formed by oxygen ions.

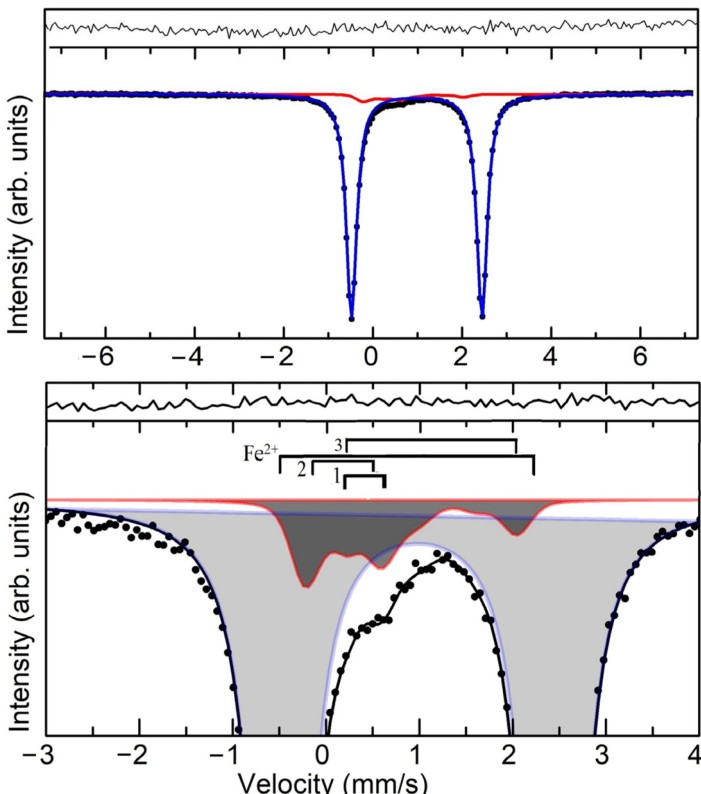

**Figure 5.** Experimental MS of the developed $LiFePO_4$ (N1) powder. Upper and lower parts—spectra are recorded with high and low speed of Doppler shift of the gamma-ray source, respectively. The best fitting results of the model spectrum are shown as a solid line.

In addition to the dominant doublet on the MS (Figure 5), in the range of velocities from −0.25 to +0.7 mm/s, low intensity broad lines (G = 0.565 mm/s) were also observed. The nature of these lines formation is a subject of discussion in the literature [110–113]. This small contribution is often ascribed to a lithium-deficient phase in $Li_{1-x}Fe^{2+}{}_{1-x}Fe^{3+}{}_xPO_4$ or to partial reduction of $LiFePO_4$ in an $Ar/H_2$ atmosphere, leading to the formation of amorphous impurity phases such as $FePO_4$ and/or $Fe_2P$ obtained by high-temperature annealing in a partially reducing atmosphere of $Ar/H_2$ [111,112]. Based on the results of [113], it can be argued that the $Fe^{2+}$ and No. 3 lines observed on the $LiFePO_4$ MS (Figure 5) with the HFI parameters IS = 0.981 (1) and QS =2.928 (1) mm/s IS= 1.156 (0.04) mm/s and QS = 1.818 (0.028) mm/s, belong to $LiFePO_4$ of the olivine type and $FePO_4$ in the amorphous state, respectively. Comparison of the [114–116] results indicates the absence of any ferromagnetic or ferrimagnetic impurities, such as $Fe_2O_3$, in the samples, at least in the samples with carbon-coated crystallites. The similarity of valence and local coordination states of $Fe^{2+}$ ions in glasses and in $LiFePO_4$ crystals may be the reasons for the easy formation of $LiFePO_4$ crystals during the crystallization of lithium iron phosphate glasses (see [117] and references therein). Therefore, the following mechanism of crystallization in glasses was considered (see [118] and references therein): first, $LiFePO_4$ crystals are formed in glasses with a high content of $Fe^{2+}$ ions, after which $Li_3Fe_2(PO_4)_3$ crystals appear in the remaining glass phase enriched with $Fe^{3+}$ ions.

Thus, the following conclusions can be drawn:

In the initial equilibrium compositions of $LiFePO_4$, the $Fe^{3+}$ content in the samples is much higher than the value that the electrochemical decrease in the Li content could provide. In the developed sample, it was at least 6–8%. The amount of $Fe^{3+}$ in the literature varies from 2% to 30% [18,51,54–56].

A ferric compound, most likely in the form of $Li_3Fe_2(PO_4)_3$, is found in TEM images (Figure 3) in the form of nanocrystallites on the surface of $LiFePO_4$ particles in both sets of—those synthesized and industrial ones. This agrees with the results obtained in [52,53,57].

For high cyclability and low sample resistance, it is necessary to have an optimal amount of $Fe^{3+}$ ferric compounds, which appear as by-products of $LiFePO_4$ synthesis. As can be seen from Table 2, the $Fe^{3+}$ content of about 2% will be insufficient. The values of 5–8% will be optimal for the capacity value [18,56,119] and cycling; significantly larger contents will be excessive [18,54–56] due to comparability of $Fe^{3+}$ concentration with the total content of Fe ions.

These results, together with the results of the RS studies, demonstrate that the surface of the developed powder is a high-quality, low-resistance and mechanically strong ferric-graphite-graphene composite with inclusions of the $Li_3Fe_2(PO_4)_3$ ferric ($Fe^{3+}$) compound with crystallite sizes < 10 nm, which increases cyclability compared to industrial cathodes.

### 5.2. Ageing Mechanisms of LiFePO$_4$ and Test Cathode Study Using XRD and EDX

Degradation of $LiFePO_4$ crystallites is only a part of the degradation of the cathode and of the battery as a whole [120–123]. The batteries were studied in the post-mortem state [120]; the possibility of using characterization techniques with spatial resolution from Å to mm-cm was discussed in [121]. The effect of stress factors (time, temperature and state-of-charge) on battery degradation during long-term testing up to 44 months was shown in [122]; the degradation mechanisms were classified into three levels—atomic, interface and electrode scale [123]. In [124] a review of manufacturer-provided characteristics of Li-ion batteries was made. A Radon–Nikodym based approach, where probability density is built first and then used to average observable dynamic characteristic was developed and applied to determination of relaxation rate distribution from experimental measurements.

Degradation of electrode powders can include deterioration of the conductive carbon network near the interfaces [125] and its amorphization [126], appearance of cracks in crystallites [125] and amorphization of their surface [127], impurity atoms introduction into the working crystallites [128]. To reduce the Fe diffusion into the electrolyte, nano-carbon coatings are used [129]. A pyrrole (PPy) coating suppresses Fe dissolution and allows for extended retention of the olivine structure [127]. Modification of carbon by using ZnO [130], and Poly(styrene sulfonic acid) membranes by polymerization of aniline improves the coating and reduces its resistance [131,132].

Separately, degradation of crystallites may result from chemical and mechanical attacks by stress–corrosion and erosion–corrosion [133]. In the case of acids present in the electrolyte, such as HF [134], impurities catalyze these attacks: iron-rich phases have a lower corrosion potential relative to $LiFePO_4$, and phosphorus-rich impurity has a higher value [135]. Corrosion proceeds especially actively at the points of concentration of mechanical stresses, in places where cracks appear on the surface of crystallites [133]. Another example of such attacks is the formation of amorphous layers of $LiFePO_4(OH)$ on the surface of crystallites when powders are kept in a humid atmosphere, or due to the moisture and OH groups residual presence in batteries [136].

Thus, based on the literature analysis, we can conclude that, in general, the growth mechanism and $LiFePO_4$ crystallites degradation is a complex chemical and electrochemical process. To describe the first irreversible phase of high-temperature degradation, their explanation is combined corrosion, stress– or erosion–induced. A significant part of the degradation in temperature ranges from growth to 100 °C can be described by the Avrami–Erofe'ev reversible mechanisms [137], Ostwald ripening reaction [138] and Ostwald's rule of stages [139], provided that the cathode powder volume is preserved, for example, by excluding its components' diffusion into the electrolyte.

### 5.3. Test Cathode Aging Study Using XRD and EDX

XRD Bruker D8 Discover diffractometer was used to determine also the unit cell volume, *V* (Table 5 and Figure S2).

**Table 5.** Results of degradation of the test LiFePO$_4$ cathodes after 100-fold cycling at 1C discharge rate; the extrapolated prediction of their cycling was given above in Table 2. $\Delta Q$—capacity reduction, Fe atomic % obtained using EDX (Section S1), cell volumes before and after cycling, Vb-Va—their changes.

| | $\Delta Q$, mAh/g | EDX, Fe, % | | $\overline{L_{VXRD}}$, nm | | $V = a \times b \times c$, nm$^3$ | | Vb-Va, nm$^3$ |
|---|---|---|---|---|---|---|---|---|
| | | Before | After | Before | After | Before | After | Before–After |
| N1 | 0.7 | 13.7 | 5.47 | 190 (20) | 160 (50) | 291.183 | 291.37 | −0.09 |
| N2 | 4.0 | 6.28 | 2.98 | 220 (50) | 210 (50) | 291.376 | 290.069 | 1.303 |
| N3 | 0.4 | 7.45 | 2.92 | 119 (11) | 164 (17) | 290.723 | 290.633 | 0.09 |
| N4 | 6.2 | 11.17 | 12.5 | 800 (200) | 150 (20) | 291.184 | 290.88 | 0.304 |

Table 5 lists the results which correspond to the first aging phase characterized with a partial destruction (or stabilization) of the protective ferric-graphite-graphene layer on the surface of the crystallites. As can be seen from Figure S1, significant deviations from stoichiometry within the LiFePO$_4$ olivine structure are possible when the equality $2\,V_{Li} \approx Fe_{Li}$ is satisfied. With these deviations, the cell volume increases and, according to [54,140], for every two lithium vacancies $2\,V_{Li}$, a $Fe_{Li}$ defect arises (iron in lithium position).

Based on Table 5 and Figures S1 and S2 the following conclusions can be drawn:

According to EDX measurements, during the first aging phase, the Fe content in the N1-3 test cathodes decreases mainly due to its diffusion from the intercrystallite space into the electrolyte; an increase in the Fe content in the lowest quality sample N4 indicates the beginning of its crystallite destruction.

For the samples N1–4, a decrease in the size of crystallites is observed, and for the samples N2–4, a decrease in the volume of unit cells Vb-Va is observed, which is proportional to a decrease in the number of cycles. The latter also means a decrease in the $Fe_{Li}$ and $V_{Li}$ defect concentrations, i.e., the crystallites approach the stoichiometric composition. However, an increase in deviation from stoichiometry is observed for the developed sample N1 in the first phase of degradation.

In the experiments with incompletely discharged test cathodes, an increase in crystallite size up to 60 nm is observed (significantly smaller than in the initial powder) as the concentration of the FePO$_4$ phase increases (see Figure S2). In this case, the unit cell dimensions of the remaining crystallites decrease, i.e., they approach the stoichiometric composition. This means that smaller crystallites have a greater deviation from stoichiometry, by 5% $V_{Li}$ relative to the average value. Note that the XRD measurement of the discharged test cathodes is lengthy; therefore, the process of relaxation of partially discharged crystallites already ends as a result of the Li redistribution relaxation between crystallites in these samples [141,142].

Thus, the LiFePO$_4$ cathode degradation occurs in two stages: at the first stage, the layer on the surface of the crystallites is destroyed; at the second stage, Fe escapes into the electrolyte and onto the anode with a decrease in the size of the crystallites due to thickening of the amorphized near-surface layer. In Figure 6, this two-stage process scheme is shown; it is close to that previously proposed for describing the Ostwald ripening reaction and Ostwald's rule of LiFePO$_4$ crystallization stages [137–139]. According to these works, at high temperatures, the growth part consists of Ostwald ripening stages from the metastable state of the feedstock with free energy $\Delta G > 0$ up to the equilibrium state with zero energies. The presence of a local electrochemical potential must also be included in the height of the barriers.

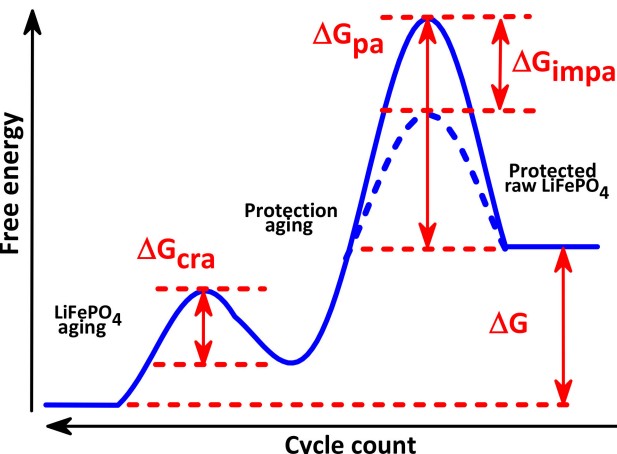

**Figure 6.** Scheme of two-stage LiFePO$_4$ cathode degradation. Here, the free energy in the metastable state is $\Delta$G at the battery operating temperatures. With an increased number of cycles, the protective ferric-graphite-graphene layer is destroyed with the corrosion activation energy $\Delta$G$_{pa}$; a decrease in this activation energy by $\Delta$G$_{impa}$ occurs in the presence of impurities that catalyze the corrosion processes. The second stage is degradation with activation energy of $\Delta$G$_{cra}$ of unprotected crystallite destruction into an amorphous phase.

## 6. The Galvanostatic Measurements of LiFePO$_4$ Test Cathodes

The main target quantitative parameters of the electrodes are: rate capability $Q(t)$ and capacity $Q_0$, limit value at charging time $t\to\infty$. These parameters are actively used in the development of electrodes [143,144], batteries [145,146] and supercapacitors [147,148] to assess the quality of crystallites [149] and their carbon coatings [150,151] in studies of degradation of powders and batteries in general [120–123]. In addition to these parameters, three characteristic discharge/charge times associated with RC electrical ($\tau_{el}$), diffusion relaxation ($\tau_d$) and electrochemical reaction at the electrode/electrolyte interface ($\tau_c$) are also important [152]. In turn, the first two consist of 3 components each, so we have 7 characteristic times in total. It was shown in [99] that the crystallite shape engineering task aiming to optimize the rate capability and increase the cathode capacity can be divided into two subtasks: 1. Achieving a large rate capability (and capacity) at big times or increasing the rate capability at small times. 2. Decreasing characteristic discharge/charge times to increase the rate capability at small times, which can be partially solved by improving the quality of their coating.

To develop the analytical dependence $Q(t)$ and use it to describe the results of galvanostatic measurements of test cathodes, it is necessary to establish a hierarchy among these 7 relaxation times of electrochemical charge exchange. To do this, it is necessary to determine the value of the specific interfacial area in the electrodes [153], taking into account the anisotropic size distribution of crystallites.

### 6.1. Test Cathode Aging Study Using XRD and EDX

Let us define the term "specific interfacial area" as $a_s = S/V$, where $S$ is the total area of projections of the test cathode crystallites onto the (010) plane, and $V$ is their total volume. It was shown in [98,99] that the combined use of the results of TEM and XRD measurements makes it possible to determine these parameters using the anisotropic size distribution of LIFePO$_4$ powder crystallites, which is described by a 3-dimensional lognormal function:

$$f(\overline{L}) = \frac{1}{L_1 L_2 L_3 \sqrt{(2\pi)^3 \det K}} \exp\left[-\frac{1}{2}\left(\ln\overline{L} - \ln\overline{\overline{L}}\right)^T \overline{\Lambda}^{-1}\left(\ln\overline{L} - \ln\overline{\overline{L}}\right)\right], \qquad (1)$$

where $\overline{L} = \begin{bmatrix} L_1 \\ L_2 \\ L_3 \end{bmatrix}$—crystallite sizes, $\overline{\overline{L}} = \begin{bmatrix} \overline{L_1} \\ \overline{L_2} \\ \overline{L_3} \end{bmatrix}$—their means,

$\overline{\Lambda} = \begin{bmatrix} \sigma_1^2 & r_{12}\sigma_1\sigma_2 & r_{13}\sigma_1\sigma_3 \\ r_{21}\sigma_2\sigma_1 & \sigma_2^2 & r_{23}\sigma_2\sigma_3 \\ r_{31}\sigma_3\sigma_1 & r_{32}\sigma_3\sigma_2 & \sigma_3^2 \end{bmatrix}$—correlation moment matrix, and the product, $Cov_{ik} = r_{ik}\sigma_i\sigma_k$—covariances, $r_{ik}$—correlation coefficients between the i-th and k-th anisotropic distributions with possible values from 0 to 1 [154], excluding negative values. Table 6 shows the parameters of the developed samples N1 and N2, described earlier in [99].

**Table 6.** Parameters of N1 and N2 samples. Column 1—parameters of crystallites, average size $\overline{L}_1$ (nm) and variance $\sigma_1$ of Lognormal distribution along the [010] axis, etc. and columns 2, 3—parameters for [100], [001] axes. Columns 4–6 are the correlation coefficients, while 7 are their average values. Column 8 shows the total area $S$ of the cross sections of crystallites on the (010) plane, 9 shows the diffusion coefficients, 10 is the electrical relaxation time (see Section 6.2 below).

| | 1 | 2 | 3 | 4 | 5 | 6 | 7 | 8 | 9 | 10 |
|---|---|---|---|---|---|---|---|---|---|---|
| | $\overline{L}_1, \sigma_1$ | $\overline{L}_2, \sigma_2$ | $\overline{L}_3, \sigma_3$ | $r_{12}$ | $r_{13}$ | $r_{23}$ | $\overline{r}$ | $a_s$, m² | $D$, nm²/s | $\tau_{el}$, s |
| N1 | 60, 0.41 | 49, 0.40 | 72, 0.38 | 0.87 | 0.64 | 0.73 | 0.75 | $2.0 \times 10^7$ | 0.16 (0.4) | 8 |
| N2 | 92, 0.43 | 108, 0.41 | 160, 0.35 | 0.72 | 0.56 | 0.53 | 0.60 | $3.1 \times 10^7$ | 0.3–2.1 (0.4) | 20 |

The calculation of $a_s$ is performed through the following steps:

- using weight (0.015 g) of the initial amount in the LiFePO$_4$ powder sample and its pycnometric density (3.6 g/cm³), the total volume of all crystallites in the cathode is calculated (V = $4.2 \times 10^{18}$ nm³),
- the average crystallite volume is calculated (Mathematica 12 notation):

$$\overline{v_{pr}} = \textbf{Total}\left[\left(\overline{f} \circ \overline{v}\right), 3\right], \tag{2}$$

where $\overline{v}$—3-dimensional N-bit matrix of particle volumes, each element of which for ellipsoid particles has a volume $\frac{\pi}{6} L_{in} * L_{jn} * L_{kn}$. Index $n$ runs over values from 1 to N, while $L_{in}, L_{jn}$ and $L_{kn}$ are the sizes of crystallites along the [010], [100] and [001] axes, respectively; $\overline{f}$ is the discretization of function (1) normalized to 1 in the form of a 3-dimensional $N$-bit matrix, each element of which means the probability of occurrence of the crystallites with the corresponding sizes. The **Total** operator means the matrix elements product and all products summation:

- dividing $V$ by $\overline{v_{pr}}$ we obtain the number of particles $N_{ct}$ in the cathode and the $S$ value by calculating an equation similar to (2). Instead of $\overline{v}$ it uses the matrix $\overline{s}$—particle area projections onto the (010) plane, and the program line is as follows:

$$S = N_{ct} \textbf{Total}\left[\left(\overline{f} \circ \overline{s}\right), 3\right], \tag{3}$$

- as a result, for the developed powder, we obtain the number of crystallites in the cathode $N_{ct} = 4.1 \times 10^{12}$; the total areas $S = 8.3 \times 10^{16}$ nm² and the specific interfacial area $a_s \approx 2 \times 10^7$ m$^{-1}$ can be calculated. The results for 2 samples are shown in Table 6.

Since we have limited ourselves to powder improvement technology, electrochemical tests were carried out by fabricating thin test cathodes using a three-electrode cell [99]. In this case, only 3 out of 7 characteristic times will remain: diffusion relaxation $\tau_d$ along the crystallite [010] axis columns, RC electric $\tau_{el}$ associated with the coating of crystallites, and the response time to the electrode/electrolyte interface $t_c$. To estimate the latter, we

use the value of specific interfacial area $a_s$ obtained above, as well as the analyses given in [152–154], quantitative calculations and the following expression:

$$t_c = \frac{F \varepsilon c_e}{(1 - t_+^0)\left|a_s i_0 exp\left(\frac{\alpha_c F}{RT}\eta\right)\right|},$$ 
(4)

where Faraday's constant $F = 96{,}487$ C·mol$^{-1}$, porosity $\varepsilon = 0.3$, electrolyte concentration $c_e = 1000$ mol·m$^{-3}$, transference number $t_+^0 = 0.4$, cathodic transfer coefficient $\alpha_c = 0.5$ taken from [155–157], and the values $a_s = 2 \times 10^7$ m$^{-1}$, surface overpotential $\eta = 0.1$ V and reference exchange current density $= 1.5$ A/m$^2$ are obtained from the developed sample measurements. Substituting numerical values into (4), we obtain $t_c = 0.5$ s. Thus, comparing the value of $t_c$ with those obtained in [155], we are convinced that in our sample with 8 µm cathode thickness, its value is indeed the minimum in the hierarchy of electrochemical charge exchange relaxation times, which allows simplifying the $Q(t)$ analytical dependence, leaving only the parameters of $f(\overline{L})$ and $\tau_d$, $\tau_{el}$.

*6.2. Q(t) Dependence on Crystallite Parameters, Lithium Diffusion Coefficient D along [010] and the Quality of Their Coating (Electrical Relaxation Time $\tau_{el}$)*

To develop an analytical model $Q(t)$, we make the following assumptions:

1. In [152,158], for some current source, for which $Q(t)$ asymptotically approaches the limit value $Q_M$ at $t \to \infty$, and at $t \to 0$ it approaches the dependence $\frac{Q_M}{2}\left(\frac{\tau}{t}\right)^{-n}$, the following empirical equation was proposed:

$$Q(t) = Q_M\left[1 - \left(\frac{\tau}{t}\right)^n\left(1 - e^{-\left(\frac{\tau}{t}\right)^{-n}}\right)\right],$$ 
(5)

   where $\tau$ is the time constant, and the exponent values $n$ are defined in [152] for batteries and supercapacitors, as 0.5 and 1.0, respectively. That is, the exponent $n$ is equal to the slope tangent of the dependence $Q(t)$ in double logarithmic coordinates at small $t$. Equation (5) can be interpreted as follows: over time, $Q(t)$ reaches its limit value $Q_M$ with probability $[1 - P]$, where $P$ is equal to the subtract in the square bracket of Equation (5) and has the meaning of the process probability not being implemented due to the limited rate [159].

2. According to [99], the crystallite is divided into columns with a cross-sectional area $dx_3 * dx_2$ along the Li diffusion direction—axis [010], along which the coordinate axis $x_1$ is directed. The crystallite rate capability $q_{cr}(t, L_1, L_2, L_3)$ is determined by integrating over the plane (010) the rate capability $q_{se}(t)$ of length M column:

$$q_{cr}(t, L_1, L_2, L_3) = \int_{-\frac{L_3}{2}}^{\frac{L_3}{2}} dx_3 \int_{-\frac{L_2}{2}}^{\frac{L_2}{2}} q_{se}(t)dx_2,$$ 
(6)

   where $L_1$, $L_2$ and $L_3$ are the crystallite dimensions along the [010], [100] and [001] axes, respectively.

3. Sequential charge carriers flow in a crystallite column through a capacitor (the model of a dense electric double layer) and an element with distributed parameters (the model of Warburg element diffusion of the stage limiting the rate of the Faradaic process). The model can be considered similar to the electrical circuit in which the capacitor and the Warburg element are series-connected [42,160].

4. The chain Figure 7 corresponds to the probabilistic equation of the sequence of events [154]:

$$q_{se}(t) = q_M(1 - P_C)(1 - P_W) = q_M\left[1 - \left(\frac{\tau_{el}}{t}\right)^1\left(1 - e^{-\left(\frac{\tau_{el}}{t}\right)^{-1}}\right)\right]\left[1 - \left(\frac{\tau_d}{t}\right)^{0.5}\left(1 - e^{-\left(\frac{\tau_d}{t}\right)^{-0.5}}\right)\right],$$ 
(7)

in which the rate capability $q_{se}(t)$ is not realized with probability $P_C$ and $P_W$ in time $t$. The dependence $\tau_d = \frac{M^2}{\pi^2 D}$ is used, where $D = $ Const, which describes the process of diffusion (desorption) from a finite size $M$ with associated boundary conditions [161].

5.    Next, similarly to Equations (2) and (3), the desired dependences are calculated:

$$Q(t, D, \tau_{el}) = Total\left[\left(\overline{f} \circ \overline{q_{cr}}\right), 3\right], \tag{8}$$

6.    Figure 8 shows the fitting of the dependence of the calculated sum results (7) on the discharge time with reference to the experimental normalization value of the rate capability at $t_{nr} = 80$ s (10 C rate), which is intermediate between the dominant contributions.

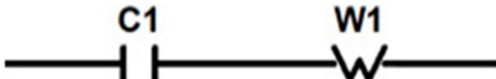

**Figure 7.** Cathode equivalent circuit of a crystallite column consisting of a capacitor C1 and Warburg element W1 series connection.

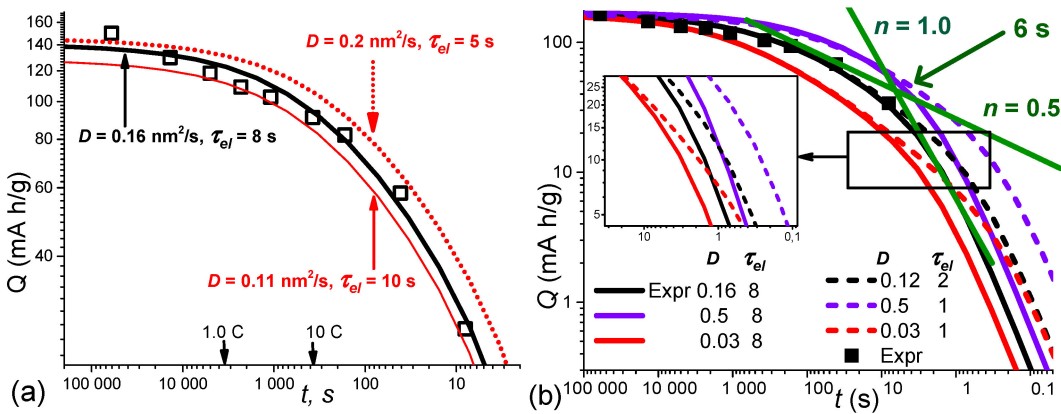

**Figure 8.** (**a**) Theoretical $Q(t, D, \tau_{el})$ dependence, Equation (8), on the discharge time for the developed powder with the steps of fitting the most optimal black curve to the experimental points. (**b**) The same optimal curve and experimental points on large scales along the axes and large values of $D$ and $\tau_{el}$ to demonstrate tilt angles. Straight lines with slopes $n$ corresponding to the Warburg element and capacitor are shown, with the intersection at the point $t = 6$ s close to the obtained value $\tau_{el} = 8$ s.

The theoretical dependence $Q(t)$ can use two relaxation times $\tau_d$ and $\tau_{el}$, which are the most important in the hierarchy of relaxation times. The procedure for fitting $Q(t, D, \tau_{el})$, expression (7), to the experimental $Q(t)$ includes the use of the distribution parameters of anisotropic crystallite sizes, as well as the normalization value $Q(t_{nr})$ at some $t_{nr}$. For the obtained value $\tau_{el} = 8$ s, the above estimate of the need to fulfill the inequality $\tau_{el} < t_c = 0.5$ s is performed with a large margin. As can be seen from Tables 2 and 6, the crystallite average sizes along the [010] axis of the developed sample N 1 are reduced by 2–3 times relative to the rest. At the same time, it is significant that the crystallite average volume is even more reduced, in particular, by a factor of 30 compared to sample N4. This fact also indicates the high quality of the developed powder, since a decrease in the crystallite volume should obviously reduce the crystallite lifetime (cycling).

*6.3. Q(t) Calculation in the Ranges of D, $\tau_{el}$ u $\overline{r}$, Close to the Values of the Developed LiFePO$_4$ Powder to Assess the Possibility of Improving Technologies*

As can be seen from Figure 8b, a 25% decrease in $D$ and a decrease in $\tau_{el}$ by a factor of 4 have practically no effect on the value of $Q(t)$ in the practically important range of

discharge rates up to 50 C rate; the latter corresponds to the rightmost experimental point. From Figure 8b, it is also seen (purple solid and dotted curves), that an obvious way to improve the powders would be technologies aimed at increasing $D$ by a factor of three and reducing $\tau_{el}$ to extremely small values equal to $t_c = 0.5$ s.

In [99 + SI] a significant number of different calculations are presented in wide ranges of anisotropic distribution parameters of crystallite sizes, which are in agreement with known experimental results. Below, the dependence of $Q(t)$ on the values of the correlation coefficients of Equation (1) (the correlation between the crystallite linear sizes) will be described.

It should be noted that experimental determination of the correlation coefficients is quite reliable for crystallites with plate- and bar-like shapes, using SEM or TEM microscopy. The largest face of the crystallites is located in the object microscope plane and, in principle, no XRD measurement is required. However, there frequently are more complicated situations, with less anisotropy (<5-fold). Here, only complementary XRD and TEM measurements are possible [98]. The essence of the technique is that the difference between the column averaged $\bar{L}_{i\,\mathrm{XRD}}$ and the volume averaged $\bar{L}_{i\,\mathrm{TEM}}$ can be related to the average sizes of real measurements $\bar{L}_i$ for log-normal distributions. In this case, the TEM measurement results produce a correlation cloud between the longitudinal and transverse sizes of crystallites and the corresponding marginal distribution functions [159]. Figure 9a shows such a cloud obtained by digitizing TEM images of the developed sample. Luckily, they are described by a lognormal function (normal in the ln coordinate). This allows us to divide them into 3 components along the crystallographic axes and then, using a fitting procedure, find the correlators $r_{ik}$ which are also shown in Table 6. The procedure is described in [99] and implies using the correlation between the longitudinal and transverse particle sizes (see Figure 9a) $r_{bs}$ as a trial to obtain $r_{ik}$. The program fits the row-by-row sum of the 3-D matrix $\bar{f}$ to the corresponding 3-D marginal size distributions along the crystallographic axes. Figure 9a shows the ordering of points, which is due to the discreteness of the particle size digitization, as well as the coincidence of some particles in sizes.

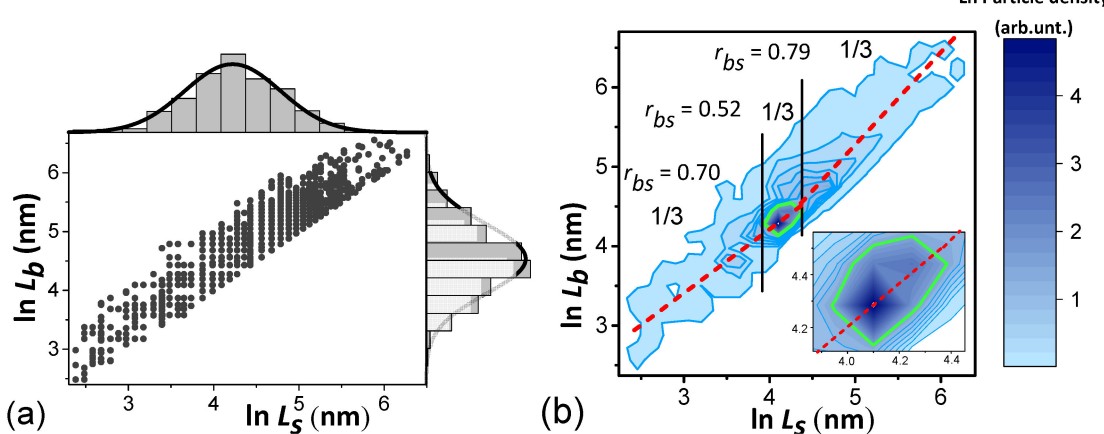

**Figure 9.** (**a**) Correlation cloud of transverse and longitudinal sizes $L_s$ and $L_b$ of LiFePO$_4$ for 4326 particles and the corresponding marginal functions of their distributions fitted with Lognormal functions. (**b**) Particle-density isolines map divided into parts by vertical straight lines with small, intermediate and large particles 1/3 of their total; red dotted lines—smoothed average values. The correlation coefficients of these parts are indicated; the inset shows an enlarged part near the particle density maximum.

Figure 9b shows a crystallite density distribution contour map. To take the correlations into account, the cloud can be discretized into sections and correlations calculated for each section. An example for 3 sections is shown in Figure 9b. However, to reduce the error in determining $r_{ik}$, it is necessary to increase the number of digitized particles, which will significantly increase the processing time, considering the overlap of the particles in TEM

images [98]. On the other hand, the difference between the $r_{ik}$ values the studied samples are relatively small, no more than 3 tenths; therefore, to determine the dependence of $Q(t)$ on this parameter, we use the $\bar{r}$ average value in a wider range from 0 to 1. The calculation results shown in Figure 10 are carried out using the average value $\bar{r}$ indicated in Table 6.

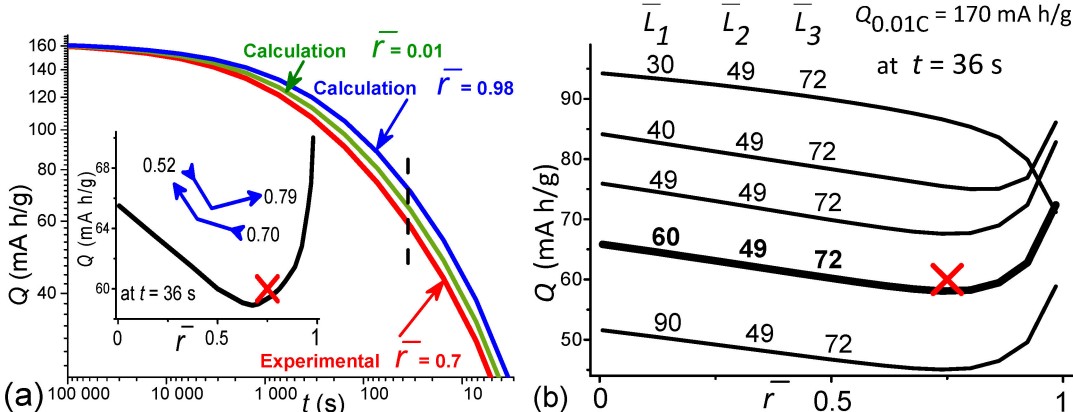

**Figure 10.** (**a**) $Q(t)$ dependences on the mean values of the correlation coefficients $\bar{r}$ In the inset, $Q(t)$ dependence on $\bar{r}$ at $t = 36$ s (marked by a vertical dotted line); the red cross marks the sample N1 $\bar{r}$ value. A scheme of $\bar{r}$ changes during measurements from small to large $t$ is shown. (**b**) $Q(t)$ dependences on $\bar{r}$ normalized to $Q_{0.01}$ at $t = 36$ s for the indicated average crystallite sizes; the thick curve and the red cross are for N1 sample.

As can be seen from Figure 10b, the dependence of $Q(t)$ on $\bar{r}$ in the range of short discharge times (high power) is nonmonotonic for the developed sample. Note that it satisfies the inequality $\bar{L}_1 > \bar{L}_2$, and $\bar{L}_1 \sim \bar{L}_3$. To increase $Q(t)$ in the region of small $t$, when these relations are satisfied, it is necessary to increase $\bar{r}$, in other words, to reduce the cloud width in Figure 9b. As can be seen in the inset to Figure 10a, this can be expected to be quite realistic, since the $\bar{r}$ achieved value is on the threshold of its sharp rise. However, when developing a powder with particle sizes $\bar{L}_1 < \bar{L}_2$, $\bar{L}_3$, the $Q(t)$ dependence decreases as $\bar{r}$ decreases; i.e., it is necessary to strive for an increase in the cloud width.

To discuss the mechanisms of the discovered $Q(t)$ dependence on $\bar{r}$, it is necessary to pay attention to the nonmonotonic change in the $r_{bs}$ value with increasing particle sizes observed in Figure 9b. The inset to Figure 10a shows a scheme according to which Li leaves small particles with $r_{bs} = 0.70$ at small discharge $t$ (high currents), and then, as the time $t$ increases, Li additionally leaves medium-sized particles with $r_{bs} = 0.52$, and finally, Li leaves large particles with $r_{bs} = 0.79$. Since $Q(t)$ is strongly dependent on the particle size, all these factors together may possibly lead to the observed dependence of $Q(t)$ on $\bar{r}$.

Thus, as shown in Figure 8b, to improve the developed powder technology, it is necessary to increase the lithium diffusion coefficient $D$ in LiFePO$_4$ crystallites by up to three times. In this case, it is necessary to improve their coatings quality from reducing the electrical relaxation time $\tau_{el}$ to the electrochemical reaction duration at the electrode/electrolyte interface $\tau_c = 0.5$ s. For an additional increase in $Q(t)$ (and power) in the region of short recharge times, it is necessary to optimize the correlation coefficients between the crystallites anisotropic sizes. They can be controlled by the cloud width of the correlations between the transverse and longitudinal dimensions of TEM particle images (see Figure 9b).

## 7. Conclusions

1. One-pot synthesis has been developed for LiFePO$_4$ powders with impurity phase content of less than 0.1%, with 2–3 times smaller crystallites along the [010] axis, with 2–3 times greater cycling compared to the industrial samples, and with the particles covered by a mechanically strong, low-resistance ferric-graphite-graphene composite protective layer with inclusions of ferric (Fe$^{3+}$) compound particles 5–10 nm

in size. The ordered carbon shell thickness reaches 5 nm, and the amorphous shell is up to 20 nm.

2. To detect impurity crystallite phases, SXRD was used, since conventional XRD is less sensitive due to the lower intensity of laboratory sources compared to the synchrotron.

3. Control of adipic acid and polyvinyl alcohol concentrations and use of multistage annealing modes makes it possible to control the coating quality. The composite layer improves cyclability compared to industrial cathodes.

4. The role of ferric $Fe^{3+}$ compounds:

- the content of ferric $Fe^{3+}$ compounds is much higher (at least 6–8%) than the value expected from the electrochemical decrease in the Li content. The amount of $Fe^{3+}$ reported in the literature varies from 2% to 30% (Table 7).

- in a controlled way, $Fe^{3+}$ compounds can be formed on the surface when the volatile components are not completely removed during $LiFePO_4$ synthesis from an intermediate low-temperature amorphous phase.

- To obtain highly cyclable and low resistant samples, it is necessary to have some optimal amount of the $Fe^{3+}$ ferric compounds, which appear as $LiFePO_4$ synthesis by-products. EDX studies of the tested cathode show that the total number of Fe atoms is reduced compared to the original samples. We have not detected $Fe_2O_3$, but it was observed in other technologies.

- According to the corrosion degradation model, an increase in the cycle number leads to a decrease in the ferric $Fe^{3+}$ compounds content on the surface of crystallites. These compounds play a certain sacrificial role [162,163], disappearing as the cathode resource is exhausted, and impurity phases can play the role of a catalyst for this breakdown. However, some of them, such as iron phosphide, weaken the catalytic activity. The degradation occurs in two stages: at the first stage, the layer on the crystallite surface is destroyed, and at the second stage, Fe escapes into the electrolyte and onto the anode with a decrease in the crystallite size due to increasing amorphization of the near-surface layer of crystallites.

**Table 7.** Parameters of LiFePO4 powders: developed, studied industrial and described in the references, as well as comments on the comparison procedure.

| | Developed | Industrial | From References |
|---|---|---|---|
| Growth T, °C | Two steps 400 °C, 670 °C | unknown | [1] 400–800; [3] 700; [5,18] 650; [6] 550; [8] 810; [19,22] 650–700; [24] 670; [33] 550–800; [37] 550; [49,55] 700; [51] 600. |
| Technology | one-pot liquid-phase | unknown | [143] More than 10 types, main: Solvothermal, Hydrothermal, Stripping synthesis, Sol-gel. |
| Protected layer | ferric-graphite-graphene | mostly ferric-graphite | [143] More than 30 types, main: [14] different carbon; [15] nanocarbons; [16–18] graphene; [19–21] sucrose; [22,23] glucose; [24–26] adipic acid; [27,28] polyvinyl alcohol; [29,30] polymeric additive; [31] ferrocene. |
| Q, mAh/g | 0.1 C 151 10 C 82 | 128–163 72–80 | [151] commercial 121–160, best MWCNT, essentially mixed; [143] capacity growth of commercial powder from 160 to 208 mAh/g, more than theoretical 170 mAh/g (*) |
| Particle sizes, nm | $\overline{L}_{V[100]}$ 66 $\overline{L}_{V[010]}$ 82 $\overline{L}_{V[001]}$ 89 | 141–230 131–261 165–242 | [1] 300–7000; [22] 240–3000; [90] 180–300; [118] 500–30,200; [136] 120; [138] 95–280; [141] 60–1000; [144] 40–500; [149] 20–140; [151] 90–300; [164] 30–158. (**) |
| Cell volumes, $^0$A | | | [22] 290.63–290.94; [36] 291.08–292.07; [88] 289.7–291.2; [90] 291.3–292.3; [114] 291.33–291.63; [118] 289.8–291.9. |

**Table 7.** *Cont.*

| | Developed | Industrial | From References |
|---|---|---|---|
| Cycling | 3500 | 800–1000, 5000 with ferric impurity | [121] 50–600; [143] 50–1000 with different capacity retentions 92–100% (**) |
| BET, $m^2/g$ | 12.5 | 9.4–13.4 | [1] 1–20; [6] 32–66; [16] 49.3–59.4; [19] 49.6; [121] 15.5; [132] 50; [136] 19–25; [164] 35. |
| $D$, $nm^2/s$ | 0.12 | 0.25–0.45 | [3] 2; [6] 2–3; [11,16] 0.01–1; [12] 4.9–7.2; [16] 3.4–8.8, 1.8–1000; [22] 109; [36] 1.2–8.2; [51] 900–2400; [119] 7400–42,000; [128] 0.83–27.3; [149] 1; [151] 0.67–11.7; [165] 1–100; [166] 0.01–10. (***) |
| $\tau_{el}$, s | 8 | 3–30 | [152] 3–200 |
| $t_c$, s | 0.5 | 0.5 | [155] 0.1 to >100 |
| $Fe^{3+}$, % | 4 | 2–8 | [36] 0.28–0.39; [51] 9–17; [52] 5.16; [54] 2–12; [56] 5–26; [111] 7; [117] 7–25; [119] 5.4–17; [140] 1.13–17 |

(*) The identification of commercial trends also requires an analysis of market feasibility information, or its public accumulation. An example would be NREL chart www.nrel.gov/pv/cell-efficiency.html (accessed on 14 November 2022) about laboratory solar cells and separately about industrial modules. (**) Difficult to compare because there are no universal certification requirements for measurements. (***) According to [165] using geometrical area of electrode, BET area, particle spherical surface etc. might distort the $D$ values. The used dimension $nm^2/s$ is more descriptive in relation to particle sizes [99].

5. Galvanostatic studies of the N1 sample test cathodes were carried out in 3 stages with an assessment of the possibility to further improve the technology.

    5.1 To develop a theoretical dependence $Q(t)$ that takes into account the 3D lognormal crystallite size distribution $f(\overline{L})$, the response time of the electrode/electrolyte interface $t_c$ is estimated using the specific interfacial area in the electrodes $a_s = S/V$, where $S$ is the total the projected area of the test cathode crystallites on the (010) plane, and $V$ is their total volume. The value $t_c = 0.5$ s is obtained.

    5.2 Comparing the $t_c$ value with those obtained in [155], it appears to be a minimal one in the hierarchy of relaxation times of electrochemical charge exchange. This makes it possible to simplify the theoretical equation for the $Q(t)$ dependence on the $f(\overline{L})$ parameters. Fitting the theoretical dependence to the experimental data gives the value of the Li diffusion coefficient, $D = 0.12$ $nm^2/s$. The value of $\tau_{el} = 8$ s satisfies the inequality $\tau_{el} > t_c = 0.5$ s

    5.3 $Q(t)$ calculations in the ranges of diffusion coefficients $D$, electrical relaxation times $\tau_{el}$, and correlation coefficients $\overline{r}$ close to the values characteristic of the developed $LiFePO_4$ powder show that a decrease in $D$ by 25% and a decrease in $\tau_{el}$ by a factor of 4 has practically no effect on the $Q(t)$ value in the practically important range of discharge rates up to 50 C. Improving the powder technology should be aimed at increasing $D$ three times and reducing $\tau_{el}$ to extremely small values closer to $t_c = 0.5$ s. For an additional increase in $Q(t)$ (and power) in the short recharge time region, it is necessary to optimize the values of the correlation coefficients between the anisotropic crystallite sizes.

6. Table 7 summarizes the obtained parameters and compares them with the known ones, taking into account comments on them.

**Supplementary Materials:** The following supporting information can be downloaded at: https://www.mdpi.com/article/10.3390/en16031551/s1, Figure S1: The dependence of the LiFePO$_4$ orthorhombic olivine structure unit cell volume on the values of Li deficiency and Fe excess; Figure S2: The results of XRD measurements of N1 test cathode depending on its discharge degree; Section S1: SEM and EDX studies of samples before and after test cathodes 100-fold cycling made from developed and commercial powders.

**Author Contributions:** D.A., powder synthesis, methodology; A.B., conceptualization, writing, software; A.K., MS investigation, analysis; A.N., SEM investigation, analysis; E.E., data curation, original draft preparation; A.U., electrochemistry investigation, analysis; I.K., XRD investigation, analysis; V.L., RS investigation, analysis; M.T., TEM investigation, analysis; E.T., data analysis, validation. All authors have read and agreed to the published version of the manuscript.

**Funding:** This research received no external funding.

**Data Availability Statement:** Not applicable.

**Acknowledgments:** Sincere thanks to V.G. Malyshkin for discussing the aging mechanisms. XRD studies were performed in Resource Center of St. Petersburg State University, SXRD studies—in Synchrotron radiation source of Kurchatov Institute Research Center, SEM studies—in Federal Joint Research Center "Material science and characterization in advanced technology", TEM studies—in Omsk Regional Center of Collective Usage SB RAS.

**Conflicts of Interest:** The authors declare no conflict of interest.

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
