# Peer review of "Phase-Homogeneous LiFePO4 Powders with Crystallites Protected by Ferric-Graphite-Graphene Composite"

_energies, doi:10.3390/en16031551_

Round 1
Reviewer 1 Report
Review report:
Authors reported “The Technology and Investigations of Phase-homogeneous LiFePO4 Powders with Crystallites Protected by Ferric-graphite-graphene Composite”. The organization of this work is good, and the discussion is well organized. Nevertheless, I have some comments which are listed below.
1. The synthesis scheme is not clear, and it should be revised in a detailed way.
2. The author claimed that their work is a novel investigation, however, myriads of works related to “LiFePO4“ have been published to date. So, authors should change the way of the presentation focusing on novelty. The introduction should be improved with a paragraph describing the novelty and importance of the work.
3. The authors must carefully claim their novelty in the INTRODUCTION. In addition, the authors need to do some formatting errors that should be carefully checked and corrected in the text.
4. The source and purity of all chemicals used should be specified. Authors should be looked at into below suggested references and can cite and take references regarding the “Source and Purity issues”: “Dalton Trans., 47 (2018), pp. 15545-15554”, “J. Energy Storage, 31 (2020), Article 101619”, “New J. Chem., 2018, 42, 19971-19978”, which references should be cited in your revised manuscript for better understanding.
5. A summary of key improvements compared to findings in the literature [provide a couple of references to indicate key improvements].
6. Please provide the comparison table, which you need to prove that your material is superior to previously reported literature.
7. The authors should add some literature descriptions to make the manuscript more convincing. I would like to suggest the authors cite the following relevant articles to enhance the background,; “J. Energy Storage, 32, 2020, 101988”; “J. Colloid Interface Sci., 609 (2022), pp. 434-446”, “RSC Adv., 9 (2019), pp. 1115-1122”, pp. 115695”, “Energies, 11 (2018), pp. 3285”, “Nanomaterials, 12 (2022), pp. 3187”
8. The reviewer also suggests that authors get professional English services to correct the grammatical error and refine the expressions in the body of the manuscript.
9. Authors should be trimmed/condensed the ‘Abstract’ and ‘Conclusion’ sections in the revised manuscript. Please keep highlights of the whole manuscript in both sections.
Author Response
Manuscript ID: energies-2156538. Title "The Technology and Investigations of Phase-homogeneous LiFePO4 Powders with Crystallites Protected by Ferric-graphite-graphene Composite".
Review report comments, answers and author changes, responses:
Review report 1:
- The synthesis scheme is not clear, and it should be revised in a detailed way.
Changes:
The synthesis scheme was significantly revised and expanded, text corrections noted in the manuscript.
- The author claimed that their work is a novel investigation, however, myriads of works related to "LiFePO4" have been published to date. So, authors should change the way of the presentation focusing on novelty. The introduction should be improved with a paragraph describing the novelty and importance of the work.
Changes:
Novelty descriptions were added in 6 text places, text corrections noted in the manuscript.
- The authors must carefully claim their novelty in the INTRODUCTION. In addition, the authors need to do some formatting errors that should be carefully checked and corrected in the text.
Changes:
Novelty were added in the INTRODUCTION, text corrections noted in the manuscript.
The text was checked and errors corrected.
- The source and purity of all chemicals used should be specified. Authors should be looked at into below suggested references and can cite and take references regarding the "Source and Purity issues": "Dalton Trans., 47 (2018), pp. 15545-15554", "J. Energy Storage, 31 (2020), Article 101619", "New J. Chem., 2018, 42, 19971-19978", which references should be cited in your revised manuscript for better understanding.
Changes:
Source and purity chemicals were specified in synthesis part. Thank you for providing examples.
- A summary of key improvements compared to findings in the literature [provide a couple of references to indicate key improvements].
Changes:
All improvements and text corrections noted. Comparisons with literature sources are collected in Table 7 and begin at text line 798 .
- Please provide the comparison table, which you need to prove that your material is superior to previously reported literature.
Changes:
Comparison table was provided. Comparisons with literary sources are collected in Table 7.
- The authors should add some literature descriptions to make the manuscript more convincing. I would like to suggest the authors cite the following relevant articles to enhance the background,; "J. Energy Storage, 32, 2020, 101988"; "J. Colloid Interface Sci., 609 (2022), pp. 434-446", "RSC Adv., 9 (2019), pp. 1115-1122", pp. 115695", "Energies, 11 (2018), pp. 3285", "Nanomaterials, 12 (2022), pp. 3187"
Changes:
These articles have been used. Thank you very much for providing their links.
- The reviewer also suggests that authors get professional English services to correct the grammatical error and refine the expressions in the body of the manuscript.
Changes:
We have used professional English editors and carefully refined the expressions in the text.
- Authors should be trimmed/condensed the 'Abstract' and 'Conclusion' sections in the revised manuscript. Please keep highlights of the whole manuscript in both sections.
Changes:
The 'Abstract' and 'Conclusion' sections have been reduced by 20% and 25%, respectively.
Sincere thanks for the kind words and very helpful comments. We really worked with great enthusiasm to improve the text.
Reviewer 2 Report
A technology has been developed for phase‐homogeneous LiFePO4 powder with a content of impurity crystalline phases of less than 0.1% according to synchrotron diffractometry (SXRD) data. A model of the rate capability, Q(t), depending on the anisotropic distribution of crystallite sizes, including the correlation between crystallite sizes, rik, the diffusion coefficient Li along [010], D, and the electrical relaxation time, τel, has been developed.
Author Response
Manuscript ID: energies-2156538. Title "The Technology and Investigations of Phase-homogeneous LiFePO4 Powders with Crystallites Protected by Ferric-graphite-graphene Composite".
Review report comments, answers and author changes, responses:
Review report 2:
Comments and Suggestions for Authors
A technology has been developed for phase‐homogeneous LiFePO4 powder with a content of impurity crystalline phases of less than 0.1% according to synchrotron diffractometry (SXRD) data. A model of the rate capability, Q(t), depending on the anisotropic distribution of crystallite sizes, including the correlation between crystallite sizes, rik, the diffusion coefficient Li along [010], D, and the electrical relaxation time, τel, has been developed.
Changes:
Text corrections noted in the manuscript.
Sincere thanks for the kind words and very helpful comments. We really worked with great enthusiasm to improve the text.
Reviewer 3 Report
The paper is interesting, I ask for some answers to my comments before any decision. Thank you.
1- How did you anneal for 1.5 hours at 400°С in an Ar atmosphere? Did you anneal the sample by steps up to 400 C or did you wait until 400 C and then enter the sample? and why the choice of 400 and 750 and not other values.
2- Please put the details of each characterization machine at the beginning of the paper in a section called caracterization material.
3- What is the free energy unit in figure 6.
4- Please change colculation to calculation in figure 10.
5- Have you used the initial components when they arrived or have they been stored in your laboratory?
6- Did you compare your results with other works of literature, if not, please do it.
Author Response
Manuscript ID: energies-2156538. Title "The Technology and Investigations of Phase-homogeneous LiFePO4 Powders with Crystallites Protected by Ferric-graphite-graphene Composite".
Review report comments, answers and author changes, responses:
Review report 3:
1- How did you anneal for 1.5 hours at 400°С in an Ar atmosphere? Did you anneal the sample by steps up to 400 C or did you wait until 400 C and then enter the sample? and why the choice of 400 and 750 and not other values.
Changes:
The synthesis scheme was significantly revised and expanded, text corrections concerning annealing and temperatures noted in the manuscript.
2- Please put the details of each characterization machine at the beginning of the paper in a section called characterization material.
Changes:
Details of each characterization machine collected in section “2.2. Characterization techniques”.
3- What is the free energy unit in figure 6.
Answer
According to the article reference [139], free energy is a function of a system in thermodynamic equilibrium. And the barrier height is the amount of energy (in eV or Joules) that must be obtained from the temperature reservoir for a single process of decomposition of the surface layer, for example, for the detachment of the Fe3+ atom from the surface of the crystallite. The height of this barrier also includes the electrochemical potential.
Changes:
A very important remark was made on line 523 of the text
“And the presence of a local electrochemical potential must also be included in the height of the barriers.”
4- Please change colculation to calculation in figure 10.
Changes:
This correction of Figure 10 was made.
5- Have you used the initial components when they arrived or have they been stored in your laboratory?
Changes:
To clarify this important issue, the following phrase was inserted on line 133 of the text
“Chemicals were used as received from the manufacturers, additional reduction of atmospheric impact was provided at the stages of pre-drying and annealing.”
6- Did you compare your results with other works of literature, if not, please do it.
Changes:
Table 7 summarizes the obtained parameters and compares them with the known ones, taking into account comments on them.
Sincere thanks for the kind words and very helpful comments. We really worked with great enthusiasm to improve the text.
Round 2
Reviewer 1 Report
It can be accepted in its current format